# *Hafnia alvei* HA4597 Strain Reduces Food Intake and Body Weight Gain and Improves Body Composition, Glucose, and Lipid Metabolism in a Mouse Model of Hyperphagic Obesity

**DOI:** 10.3390/microorganisms8010035

**Published:** 2019-12-23

**Authors:** Nicolas Lucas, Romain Legrand, Camille Deroissart, Manon Dominique, Saïda Azhar, Marie-Anne Le Solliec, Fatima Léon, Jean-Claude do Rego, Pierre Déchelotte, Sergueï O. Fetissov, Grégory Lambert

**Affiliations:** 1TargEDys SA, University of Rouen Normandy, 22, Boulevard Gambetta, 76183 Rouen, CEDEX 1, France; lucasnicolas@hotmail.fr (N.L.); legrandromain@hotmail.fr (R.L.); camillederoissart@gmail.com (C.D.); mdominique@targedys.com (M.D.); saida-az@hotmail.fr (S.A.); marie-anne.le-solliec@univ-rouen.fr (M.-A.L.S.); pierre.dechelotte@chu-rouen.fr (P.D.); 2Inserm UMR1073, Nutrition, Gut and Brain Laboratory, 76183 Rouen, France; 3Institute for Research and Innovation in Biomedicine (IRIB), University of Rouen Normandy, 76183 Rouen, France; Fati.jpl@free.fr (F.L.); jean-claude.dorego@univ-rouen.fr (J.-C.d.R.); 4Animal Behavioral Platform, SCAC, University of Rouen Normandy, 76183 Rouen, France; 5Rouen University Hospital, CHU Charles Nicolle, 76183 Rouen, France; 6Inserm UMR1239, Laboratory of Neuronal and Neuroendocrine Differentiation and Communication, University of Rouen Normandy, 76130 Mont-Saint-Aignan, France

**Keywords:** obesity, food intake, microbiota, probiotics, *Hafnia alvei*, animal models

## Abstract

Use of new generation probiotics may become an integral part of the prevention and treatment strategies of obesity. The aim of the present study was to test the efficacy of a potential probiotic strain of lactic bacteria *Hafnia alvei* (*H. alvei*) HA4597™, in a mouse model of obesity characterized by both hyperphagia and diet-induced adiposity. For this purpose, 10-week-old high-fat-diet (HFD)-fed hyperphagic *ob*/*ob* male mice received a daily treatment with 1.4 × 10^10^ CFU of *H. alvei* for 38 days. Effects of *H. alvei* were compared to those of a lipase inhibitor orlistat (80 mg/kg daily) and a vehicle (NaCl 0.9%) in HFD-fed *ob/ob* mice. A control untreated group of *ob/ob* mice received the standard diet throughout the experiment. The vehicle-treated HFD group displayed increased food intake, worsening of adiposity, and glycemia. Treatment with *H. alvei* was accompanied by decreased body weight and fat-mass gain along with reduced food intake to the level of the standard-diet-fed mice. At the end of the experiment, the group treated with *H. alvei* showed a decrease of glycemia, plasma total cholesterol, and alanine aminotransferase. The orlistat-treated mice showed a lower rate of body weight gain but were hyperphagic and hyperglycemic. These results demonstrate the beneficial anti-obesity and metabolic effects of *H. alvei* HA4597™ in mice with obesity resulting from hyperphagia and diet-induced adiposity.

## 1. Introduction

According to the World Health Organization, the global prevalence of overweight (BMI ≥ 25 kg/m^2^) and obesity (BMI ≥ 30 kg/m^2^) has tripled since 1975, affecting about 39% and 13%, respectively, of adults in 2016 and about 381 million children and adolescents [1]. Considering that obesity is socially debilitating and a major risk factor for several chronic diseases, including diabetes, cardiovascular diseases, and cancer [2], the alarming situation with world ‘obesity epidemics’ urges for the search of new prevention and treatment solutions.

Current knowledge on the pathophysiology of obesity involves altered signaling to the brain from adiposity hormones, such as leptin and insulin, resulting in hyperphagia and positive energy balance [3]. Moreover, gut microorganisms have recently been implicated in the regulation of host energy metabolism by showing the associations of gut microbiota composition with healthy or obese phenotypes [4,5]. Although the molecular pathways linking gut microbiota to the brain regulation of energy balance are not yet fully understood, modulation of gut microbiota composition by supplementing beneficial to health bacteria appears as an appealing new strategy for obesity prevention and treatment. Indeed, traditional probiotic bacteria belonging to the genera of Lactobacillus, Bifidobacterium, and Enterococcus, alone or in various combinations, have been tested in humans, resulting typically in minor health-positive effects, without significant changes in food intake and body weight [6,7,8,9]. Nevertheless, existence of the functional link between gut microbiota and host regulation of appetite and energy metabolism indicates that some gut commensal bacteria can be potentially identified and developed as new-generation anti-obesity probiotics based on the understanding of their mechanisms of action [10].

We have recently demonstrated that a *Hafnia alvei* (*H. alvei*) HA4597™ strain of commensal bacteria of the Hafniaceae family (formerly Enterobacteriaceae) belonging to the order of Enterobacteriales displays anti-obesity properties in two animal models of obesity, standard-diet-fed leptin-deficient *ob/ob* and high-fat-diet (HFD)-fed wild-type (WT) mice [11]. The choice of *H. alvei* as a potential anti-obesity probiotic was based on its production of the caseinolytic protease B (ClpB) protein, which was previously identified as a conformational mimetic of α-melanocyte-stimulating hormone (α-MSH), a key anorexigenic peptide involved in the regulation of appetite [11,12].

In the present study, we aimed to test the anti-obesity efficacy of the *H. alvei* HA4597™, in a mouse model of obesity characterized by both hyperphagia and HFD-induced adiposity. In fact, such forms of obesity may most closely reflect the obesity phenotypes in many humans (compulsive eating behavior combined with hypercaloric diet and accompanied by functional leptin resistance), while *ob/ob* and HFD models used separately display some limitations. The lack of leptin in *ob/ob* mice results in strong hyperphagia underlying development of obesity even on a standard diet [13]. On the other hand, WT mice fed HFD develop diet-induced obesity (DIO), which is typically accompanied by a decreased total (but not caloric) food intake, a phenomenon which was also observed in our previous study, representing an obstacle for the evaluation of a potential *H. alvei* anorexigenic effect in palatable HFD [11]. An animal model of obesity based on HFD-fed *ob/ob* mice may represent a good alternative most closely reflecting hyperphagia and diet-induced obesity in humans. Indeed, it has been known for a long time that *ob/ob* mice display high retention of nutrient-derived energy, which is further enhanced by consumption of HFD [14]. HFD induction of adiposity is associated with a proportional increase of plasma leptin levels which do not result in decreased caloric consumption, witnessing the development of a leptin resistance, whereas feeding of HFD to *ob/ob* mice is accompanied by even further increase of their total food intake [15,16].

Thus, in the present study, we used HFD to exacerbate obesity in hyperphagic *ob/ob* mice and then treated them chronically with *H. alvei* HA4597™, to compare its effects with untreated mice. We also compared the anti-obesity effects of *H. alvei* with the drug orlistat, a lipase inhibitor used in humans for body weight control. We show here that *H. alvei* provision was able to decrease total food intake, body weight, and fat content and to partially restore other obesity-related metabolic parameters.

## 2. Materials and Methods

### 2.1. Animals and Experimental Procedure

Animal care and experimentation were done in accordance with guidelines established by the National Institutes of Health, French and European Community regulations (Official Journal of the European Community L 358, 18/12/1986) and were approved by the Local Ethical Committee of Normandy (n°5986). Six-to-seven-week-old male *ob*/*ob* mice (B6.V-Lep *ob*/*ob* JRj—*n* = 88) were purchased from Janvier Labs ( Le Genest-Saint-Isle, France), and, upon arrival, they were kept in a specialized air-conditioned animal facility (22 ± 2 °C, relative humidity 40 ± 20%, reverse 10:00 p.m. to 10:00 a.m. light/dark cycle). During acclimation to the animal facility (from day (D)_-19_ to D_-12_), mice were housed in standard holding cages (*n* = 3 per cage) with pelleted standard rodent chow (3430 standard diet, KlibaNafag, Kaiseraugst, Switzerland) and drinking water available ad libitum. After acclimation, the mice were weighed and placed in a restraint cylinder, and their body composition, including lean and fat mass, as well as body fluids, was measured, using the MiniSpec LF50 (Bruker, Rheinstetten, Germany). Then, according to body weight, the animals were randomly divided into two groups: (i) fed with a HFD (*n* = 72, 45 kcal% fat, 35 kcal% carbohydrates, 20 kcal% proteins, energy density 4.7 kcal/g, D12451, Research Diets, New Brunswick, NJ, USA) and (ii) fed with a standard diet (SD; *n* = 16, 5 kcal% fat, 75 kcal% carbohydrates, 20 kcal% proteins, energy density 3.8 kcal/g, 3430 KlibaNafag). The animals underwent a 5-day “induction” to both diets, followed by 1 week of daily handling protocol, where they were sham-treated (oral gavage of 200 µL of NaCl 0.9%, using rounded-end cannulas, Socorex, Ecublens, Switzerland) twice a day (approximatively 10:00 a.m. and 17:00 p.m.) from D_-7_ to D_0_ (Figure 1). This procedure has been found to reduce stress-related effects on the body weight in chronic studies. During the baseline period and the later treatment sessions, body weight, as well as food intake (by cage), were measured daily, before the morning gavage (at approximatively 9:30 a.m.).

### 2.2. Treatment Protocol

Toward the end of the baseline (D_0_), body composition was measured for a second time, using the MiniSpec LF50 (Bruker), and the HFD-fed animals were randomly divided into 3 subgroups (*n* = 24 in each). All animals from the same cage were affiliated to the same subgroup. From D_0_ onward, animals from each of the 3 HFD-fed groups were treated daily for 38 days, at approximatively 7:00 p.m., by intragastric gavage, with a volume of 5 mL/kg containing either: (i) *H. alvei* HA4597™ provided by Biodis (Noyant, France) for TargEDys SA (1.4 × 10^10^ CFU/day in NaCl 0.9%); (ii) orlistat 80 mg/kg/day in NaCl 0.9% (Tocris BioScience, Bristol, UK); or (iii) NaCl 0.9% as a vehicle. The standard diet (SDiet)-fed controls received the same 5 mL/kg volume of NaCl 0.9% by intragastric gavage (Figure 1).

### 2.3. Plasma Dosage 

At D_38_, the body composition of all animals was measured, using the MiniSpec LF50, and treatment groups were divided into two sub-groups, for the oral glucose tolerance test (OGTT) after overnight fasting and for the baseline plasma glucose assay. An overnight fasting, after the removal of food during D_22_ light phase, (approximatively at 17:00 p.m.) was performed on half of each group, while food remained available ad libitum for the rest of the animals. The next morning, the mice received 30% glucose solution (2 g/kg) via intragastric gavage. Blood samples (one drop from tail tip) were taken every 15 min during 2 h, for glucose assay, using a glucometer (FreeStylePapillon Vision, Abbott Diabetes Care, Oxon, UK).

Mice which didn’t undergo the fasting procedure were anesthetized right after body composition measurements on D_38_ by intraperitoneal injection of ketamine/xylazine solution (80:10 mg/kg). Terminal blood samples (approximately 0.5 mL) were taken by a puncture of abdominal aorta in tubes coated with lithium heparin, which were stored at 4 °C, until biochemical measurements of obesity-related metabolic markers using IDEXX Catalyst^®^ One technology (Catalyst^®^ Chem 17 CLIP + Catalyst^®^ TRIG, IDEXX, Laboratories, Inc., Westbrook, ME, USA). 

### 2.4. Statistical Analysis

Results, expressed as mean ± standard error of means (SEM), were analyzed, using GraphPad Prism 5.02 (GraphPad Software Inc., San Diego, CA, USA), with a *p*-value < 0.05 considered as statistically significant. Longitudinal group differences were compared by the two-way repeated measurement (RM) analysis of variance (ANOVA) with Bonferroni’s posttests. Individual differences were analyzed, using Student’s *t*-test or Mann–Whitney’s test according to normality evaluated by the D’Agostino–Pearson’s test.

## 3. Results

### 3.1. Induction of the DIO in ob/ob Mice

After five days of the DIO induction phase, HFD-fed mice presented a significant increase of body weight as compared to SDiet-fed mice (44.7 ± 0.4 g vs. 39.5 ± 0.9 g at D_-7_, *p* < 0.001), which persisted during the sham-treatment period of the DIO induction (Figure 2A). The total daily food intake and the cumulative caloric intake were increased in HFD-fed vs. SDiet-fed mice during the sham-treatment period (Figure 2B,C). At the end of the sham period, an increase of fat mass (Figure 2D) and percentage of body fat (Figure 2E) was found in HFD-fed mice. In spite of a small increase of the lean mass (Figure 2F), its body percentage was lower in HFD-fed vs. SDiet-fed mice (Figure 2G). The changes of body composition led to an overall decrease of lean/fat mass ratio in the HFD group (Figure 2H). No significant changes of the body composition were observed in the SDiet group during sham treatment.

### 3.2. Effects of Hafnia alvei HA4597™ Supplementation on Obesity-Related Parameters

#### 3.2.1. Effects on Body Weight 

HFD-fed control mice displayed increased body weight gain as compared to SDiet-fed mice (Figure 3A–C). Supplementation of *H. alvei* in HFD mice resulted in a decreased body weight gain vs. untreated HFD group, as compared by the area under curve (Figure 3C), reaching a difference of 15.3% at the end of the treatment (Figure 3B). Accordingly, the body weight gain of the *H. alvei*-treated group was 58.1% lower as compared to the difference between HFD and SDiet groups, although such decrease did not reach significance by ANOVA (Figure 3B). The orlistat-treated group showed a significant decrease in body weight gain as compared to both the HFD control and *H. alvei* groups, reaching to the level of the SDiet-fed group (Figure 3A–C).

#### 3.2.2. Effects on Body Composition

HFD increased percentage of fat mass and decreased percentage of lean mass, resulting in lower lean/fat mass ratio vs. SDiet-fed mice (Figure 4A–C). H. alvei HA4597™ treatment alleviated the increase of adiposity and loss of lean mass associated with HFD (Figure 4A,B). These changes led to a significant improvement of the lean/fat mass ratio in mice treated with *H. alvei* (Figure 4C). Orlistat-treated HFD mice displayed similar improvements of body composition to the mice treated with *H. alvei* (Figure 4A–C).

#### 3.2.3. Effects on Food Intake

Daily food intake was higher in HFD controls vs. SDiet-fed mice during the first three weeks, and then it declined to the level of the SDiet group (Figure 5A). *H. alvei* HA4597™-treated HFD-fed mice displayed similar daily food intake to SDiet-fed mice (Figure 5A). Cumulative food intake was also reduced in *H. alvei* HA4597™-treated mice as compared to HFD controls (Figure 5B,C). In the end of *H. alvei* HA4597™ treatment, at D_38_, the cumulative food intake was lower by 26.7 g vs. the HFD controls (Figure 5C). Orlistat-treated mice displayed hyperphagia throughout the treatment (Figure 5A), resulting in an increased cumulative food intake (Figure 5B,C).

#### 3.2.4. Effects on Glycemia

Ad libitum HFD-fed *H. alvei*-treated mice displayed a significant decrease, by about 1.5-fold, of the basal glucose plasma levels as compared to both the SDiet and HFD-fed controls (Figure 6A). The OGTT test after an overnight fasting showed elevated levels of fasting glycemia in HFD-fed control mice (Student’s *t*-test *p* < 0.05) and in orlistat-treated mice as compared to SDiet-fed controls (Figure 6B). After the oral glucose load, a 15 min peak of glucose was observed in all groups, without significant difference. A return to the baseline was delayed in the orlistat-treated mice as compared to both SDiet-fed and *H. alvei* HA4597™-treated mice (Figure 6B).

#### 3.2.5. Effects on Blood Lipids and Alanine Aminotransferase

Triglycerides plasma level increased, although not significantly, in HDF-fed mice as compared to SDiet, and no significant effect of *H. alvei* on plasma triglycerides was observed (Figure 6C). In HFD + Orlistat mice, triglycerides plasma level increased significantly (Figure 6C). Total cholesterol plasma level increased significantly in HFD-mice as compared to control HFD-mice, and this increase was prevented in *H. alvei*-treated HFD-mice (Figure 6D). Orlistat reduced total cholesterol even more than *H. alvei* (Figure 6D). Plasma levels of alanine aminotransferase (ALAT), an indicator of obesity-induced steatohepatitis, were increased in HFD-mice as compared to SDiet-mice; treatment with either *H. alvei* HA4597™ or orlistat prevented this ALAT increase (Figure 6E).

## 4. Discussion

The main findings of the present study demonstrate that the oral administration of the *H. alvei* HA4597™ strain alleviated adiposity and decreased food intake in a mouse model of mixed hyperphagic and DIO-induced obesity. In fact, the induction of DIO in *ob/ob* mice was already significant after five days of HFD. After 12 days, the HFD-mice had gained about 5 g more body weight than SDiet-fed mice. This trend for HFD-induced additional weight gain continued in the vehicle-treated HFD group and resulted in increased food intake and fat mass. The increased total food intake in the HFD control group was observed for about 30 days, and it was similar thereafter with the SDiet-fed group, i.e., only increased caloric intake persisted in HFD-fed *ob/ob* mice until the end of experiment, reflecting a typical feature of caloric but not total hyperphagia in most of the HFD rodent models of obesity. Thus, these metabolic and behavioral changes were an adequate background for a preclinical model of hyperphagic DIO for testing anti-obesity effects of the potential new-generation probiotic bacteria *H. alvei*.

This study showed that a daily provision of 1.4 × 10^10^ CFU of *H. alvei* HA4597™ in *ob/ob* mice with HFD-aggravated obesity decreased body weight gain and total fat mass and preserved lean mass, resulting in an improved lean/fat mass ratio. Other obesity-induced metabolic alterations, including glycemia, total cholesterol, and hepatic ALAT, were improved by *H. alvei* treatment. *H. alvei*-associated decrease in fat mass gain and beneficial metabolic effects, including glycemia, can partly be attributed to the negative energy balance resulting from a lower intake of nutrients, particularly fat and carbohydrates. This observed anorexigenic effect is likely a unique property of *H. alvei* among probiotic bacteria so far tested in animal models of obesity. In fact, although some limited anti-obesity effects were reported with other bacteria strains, none of those decreased food intake [17].

In contrast to the anorexigenic effect of *H. alvei*, orlistat, although efficient to reduce body weight in HFD-fed mice, induced a strong hyperphagic effect, which was most likely due to the decreased fat absorption in the gut. At the end of treatment period, food intake in the orlistat-treated HFD-mice was markedly increased, and, consequently, the difference between orlistat and *H. alvei* effects on the reduction of body weight gain tended to decrease. Moreover, such compensatory hyperphagia was accompanied at the end of the treatment period by increased basal and OGTT-induced plasma glucose levels. Although these orlistat-induced unwanted effects might be specific to this animal model, they suggest that *H. alvei*-induced moderate but sustained reduction of body mass gain through decreased food intake, along with beneficial metabolic effects, may represent a safer and more-efficient strategy for long-term body weight management. 

The orexigenic effect of orlistat warrants further investigation in light of the large use of this drug for the management of body weight and obesity. It has been earlier reported that some compensatory hyperphagia directed toward fatty meals may occur after some weeks of treatment of treatment with orlistat [18]. This may be related to the reduction of lipids absorption and subsequent satiety signals. Other authors reported that, after six months, craving for food occurred in patients treated with orlistat with a preference for carbohydrates, which have a high hedonic value [19]. The presence of compulsive behavior in obese patients is clearly associated with a poorer response to orlistat treatment in terms of weight loss but also of depression level [20]. Emotional eating was also predictive of a lesser weight reduction with orlistat [21]. To our knowledge, the evolution of eating behavior in patients seeking weight loss with the use of so-called “fat-binders” has not been evaluated. Thus, a therapeutic strategy based only on the reduction of fat absorption by different ways is likely to expose to compensatory and rebound effects, because the underlying maladapted eating habits and eating behaviors are not addressed, and they may even worsen with the erroneous feeling that fat malabsorption would compensate for excess calorie intake. Thus, improving satiety signaling, together with patient education, is likely to provide a better long-term effect.

The anorexigenic effect of *H. alvei* may arise from its ability to enhance satiety signaling in the host via production of the α-MSH-mimetic ClpB protein. α-MSH is a major anorexigenic signal in both the gut and the brain that also stimulates secretion of peptide YY (PYY) in the gut, one of the main satiety hormones [22]. We have recently shown that ClpB was necessary for anorexigenic and anti-obesity effects of *E.coli* K12 in *ob/ob* mice fed a standard diet [11]. Moreover, enterobacterial ClpB was able to dose-dependently stimulate secretion of PYY by rat intestinal mucosa in vitro, suggesting that its bacterial production in the gut may be involved in the PYY-mediated hormonal signaling of intestinal satiety [23]. Nutrient-stimulated bacterial growth is indeed accompanied by increased ClpB production [24], while proteins appear as the principal macronutrient increasing ClpB mRNA and protein expression [23]. Thus, since ClpB is able to stimulate PYY secretion, presence in the gut of ClpB-producing bacteria should be able to enhance post-meal satiety. 

Another anti-obesity effects of ClpB can be mediated via its systemic and central actions, including direct activation of the anorexigenic proopiomelanocortin (POMC) neurons of the hypothalamic arcuate nucleus [24]. Besides the anorexigenic effect, activation of POMC neurons ex. by leptin, leads to increased sympathetic activity and lipolysis [25,26]. Indeed, increased fat-tissue levels of hormone-sensitive lipase were found in *H. alvei*-treated *ob/ob* mice in our previous study [11]. ClpB activation of POMC neurons may also contribute to plasma glucose lowering antidiabetic effects of *H. alvei* [27].

*H. alvei* bacteria synthetize the ClpB protein with the α-MSH-like motif present in the order of Enterobacteriales, i.e., in both *H. alvei* and *E.coli*, but not in other traditional probiotic bacteria [11,28]. Such an α-MSH-like selectivity of the ClpB structure may underlie the link between a low abundance of *Enterobacteriaceae* in gut microbiota and obesity in humans [29,30]. Our recent study also found that the *ClpB* gene was depleted in gut microbiota in obese subjects [11]. Furthermore, weight loss and decreased plasma cholesterol after Roux-Y gastric bypass and sleeve gastrectomy were associated with an increased prevalence of *Enterobacteriaceae* in the microbiota of patients six months after bariatric surgery [31].

Although *H. alvei* was detected in the gut microbiome of some healthy humans, its low prevalence (1–2%) suggests that these common dairy-product-associated bacteria are of the transient type [11]. In fact, in spite of high numbers of *H. alvei* present in some cheeses [32], their ingestion by large population does not apparently lead to *H. alvei* colonization of the human gut, a situation common for other probiotics, unless they are taken early postnatally [33,34].

The limitations of this study is the use of a genetic model of hyperphagia due to the mutation of the *ob* gene, while hyperphagia in obese humans can have mainly the hedonic origin, although it is accompanied by the functional leptin deficiency [35]. We also do not know whether introducing *H. alvei* may modify the gut microbiota composition, which may in turn play a role in anti-obesity effects. This question should be addressed in future studies. 

## 5. Conclusions

In conclusion, the present study showed that a daily provision of the *H. alvei* HA4597™ strain in genetically obese and hyperphagic *ob/ob* mice with HFD-exacerbated obesity decreased body weight gain, improved body composition, decreased food intake, and ameliorated several metabolic parameters, including plasma glucose and total cholesterol levels. These data further preclinically validate the anti-obesity efficacy of *H. alvei* as a potential new-generation probiotic for appetite and body weight control in obesity characterized by hyperphagia and diet-induced alterations of glucose and lipid metabolism.

## Figures and Tables

**Figure 1 microorganisms-08-00035-f001:**
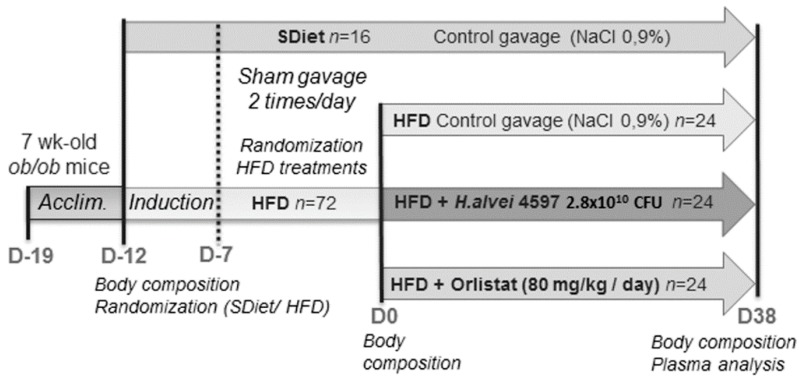
Experimental protocol of the study.

**Figure 2 microorganisms-08-00035-f002:**
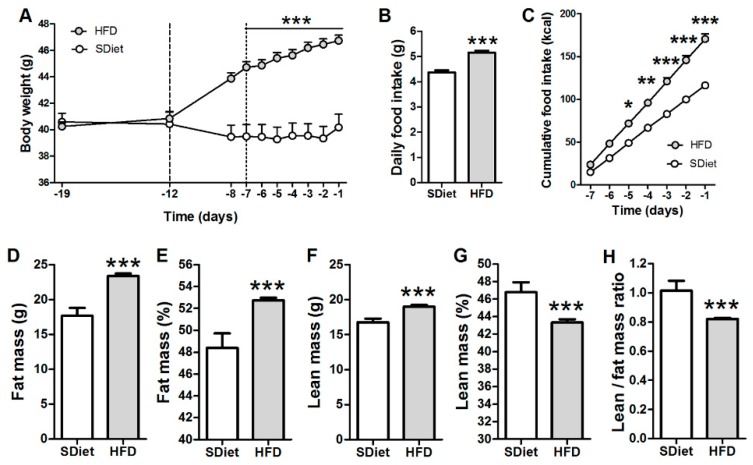
Induction of diet-induced obesity by high-fat-diet (HFD) in *ob/ob* mice. (**A**) Body weight dynamics during the pretreatment period. Vertical dotted lines define the “induction” period. Daily total (**B**) and cumulative (**C**) food intake/mouse during the sham-feeding period. Fat (**D**) and lean (**F**) mass, as well as their percentage (**E,G**) at D_0_. (**H**) Lean/fat mass ratio at D_0_. (**A,C**) Two-way RM ANOVA, both *p* < 0.0001, Bonferroni’s posttests *** *p* < 0.001, ** *p* < 0.01, * *p* < 0.05; (**B**,**D**–**H**) Mann–Whitney tests, *** *p* < 0.001, (mean ± SEM; standard diet (SDiet), *n* = 16 and HFD, *n* = 72).

**Figure 3 microorganisms-08-00035-f003:**
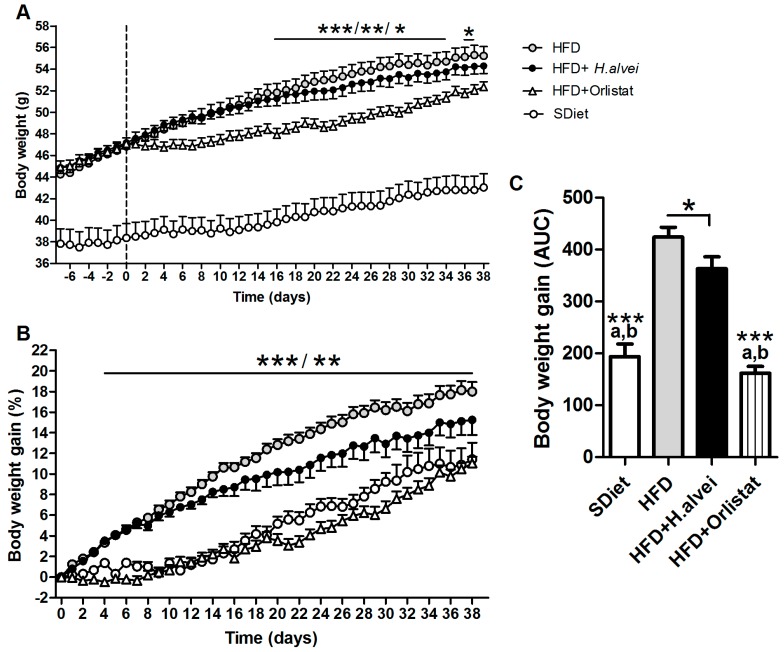
Effects of *H. alvei* HA4597™ and orlistat on body weight in high-fat-diet (HFD)-fed *ob/ob* mice as compared to standard diet (SDiet). (**A**) Body weight dynamics, vertical dashed line defines beginning of *H.alvei* or orlistat treatments (**B**) Body weight gain dynamics. (**C**) Body weight gain during treatment as an area under curve (AUC). (**A**,**B**) Two-way RM ANOVA, *p* < 0.0001, Bonferroni’s posttests, HFD vs. HFD + Orlistat, *** *p* < 0.001,** *p* < 0.01, * *p* < 0.05; **C**. ANOVA, *p* < 0.0001, Tukey’s posttests ^a^vs. HFD and ^b^vs. HFD + *H. alvei*, both *** *p* < 0.001, Student’s *t*-tests, * *p* < 0.05, (mean ± SEM; SDiet, *n* = 16, all other groups, *n* = 24).

**Figure 4 microorganisms-08-00035-f004:**
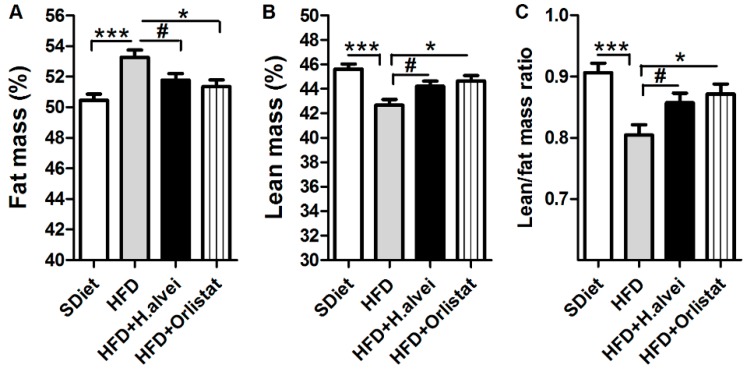
Effects of *H. alvei* HA4597™ and orlistat on body composition in high-fat-diet (HFD)-fed *ob/ob* mice as compared to standard diet (SDiet). Percentage of fat (**A**) and lean (**B**) mass, as well as lean/fat mass ratio (**C**) at the end of the treatment. (**A**,**C**) ANOVA, *p* = 0.0004, (**B**) ANOVA, *p* = 0.0002, Bonferroni’s posttests, *** *p* < 0.001, * *p* < 0.05. (**A**–**C**) Student’s *t*-tests, # *p* < 0.05, (mean ± SEM; SDiet, *n* = 16, all other groups, *n* = 24).

**Figure 5 microorganisms-08-00035-f005:**
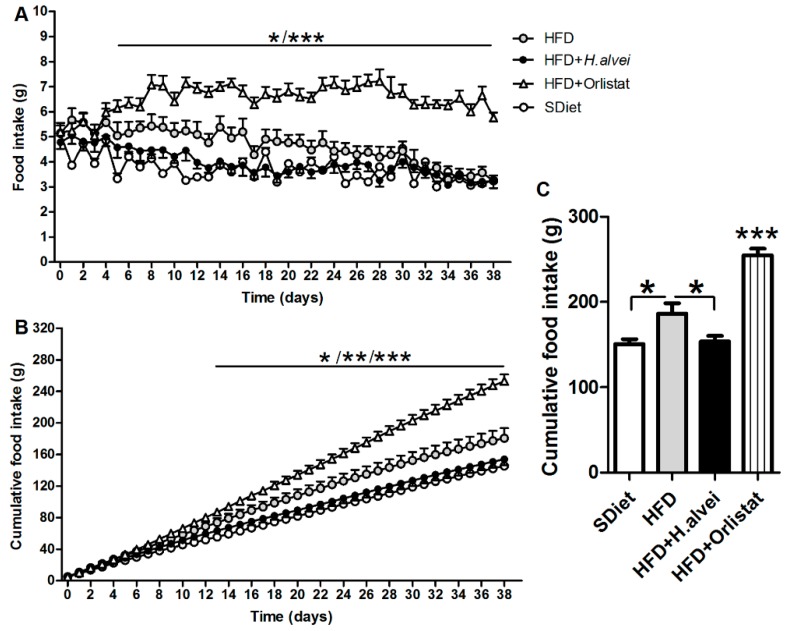
Effects of *H. alvei* HA4597™ and orlistat on food intake in high-fat-diet (HFD)-fed *ob/ob* mice as compared to standard diet (SDiet). (**A**) Dynamics of daily food intake. **(B**) Dynamics of cumulative food intake/mouse. (**C**) Mean cumulative food intake/mouse. (**A**) Two-way RM ANOVA, *p* < 0.0001, Bonferroni’s posttests, HFD vs. HFD + Orlistat. *** *p* < 0.001, * *p* < 0.05 days 8,9,14 and 19. HFD + *H. alvei* vs. HFD + Orlistat. *** *p* < 0.001, * *p* < 0.05 days 5 and 6. (**B**) Two-way RM ANOVA, *p* < 0.0001, Bonferroni’s posttests, HFD vs. HFD + Orlistat. *** *p* < 0.001; ** *p* < 0.01 days 24,25; * *p* < 0.05 days 22,23. HFD + *H. alvei* vs. HFD + Orlistat. *** *p* < 0.001, ** *p* < 0.01, days 17,18; **p* < 0.05 days 15,16. (**C**). ANOVA *p* < 0.0001, Tukey’s posttests *** *p* < 0.001, * *p* < 0.05, (mean ± SEM; SDiet, *n* = 16, all other groups, *n* = 24).

**Figure 6 microorganisms-08-00035-f006:**
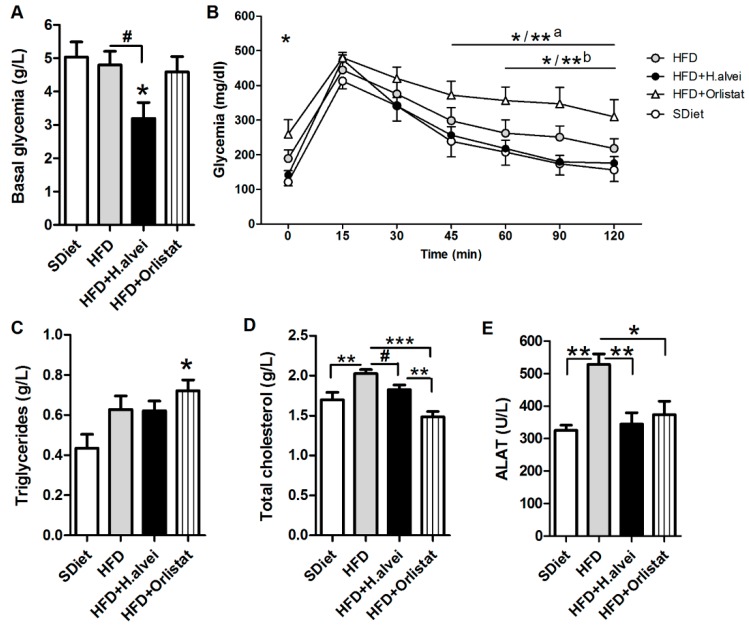
Effects of *H. alvei* HA4597™ and orlistat on glycemia and obesity-related metabolic parameters in high-fat-diet (HFD)-fed *ob/ob* mice as compared to standard diet (SDiet). (**A**) Plasma glucose levels in ad libitum feeding conditions. (**B**) OGTT after overnight fasting. Plasma levels of (**C**) triglycerides, (**D**) total cholesterol, and (**E**) alanine aminotransferase (ALAT) in ad libitum feeding conditions. (**A**). ANOVA *p* < 0.05, Tukey’s posttest vs. SDiet * *p* < 0.05. Student’s *t*-test # *p* < 0.05. (**B**) Two-way RM ANOVA *p* < 0.05, Bonferroni posttests, SDiet vs. HFD + orlistat * *p* < 0.05 and ^a^* *p* < 0.05, ** *p* < 0.01 for 90 and 120 min; HFD + *H. alvei* vs. HFD + Orlistat ^b^* *p* < 0.05, ** *p* < 0.01 for 90 min. (**C**) ANOVA *p* < 0.05, Tukey’s posttest vs. SDiet * *p* < 0.05. (**D**) ANOVA *p* < 0.001, Tukey’s posttests *** *p* < 0.001, ** *p* < 0.01, Student’s *t*-test # *p* < 0.05. (**E**) ANOVA *p* = 0.0006, Tukey’s posttests ** *p* < 0.01, * *p* < 0.05, (mean ± SEM; SDiet, *n* = 8, all other groups *n* = 12).

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
