# Peer review of "Hafnia alvei HA4597 Strain Reduces Food Intake and Body Weight Gain and Improves Body Composition, Glucose, and Lipid Metabolism in a Mouse Model of Hyperphagic Obesity"

_microorganisms, 2019, doi:10.3390/microorganisms8010035_

Round 1
Reviewer 1 Report
This is a didactic and very well-written manuscript covering a topic of great interest given the current "epidemic" of obesity.
The only mandatory revision is the necessity to report findings as mean +/- standard deviation. Using the standard error of the mean is the incorrect and somewhat misleding presentation.
Author Response
Response. We thank the Reviewer for the appreciation of our manuscript and the comments.
As the standard error (SE) is a type of standard deviation, confusion is understandable. In our manuscript we used SE of means to take in account the individual variability intrinsic to animal data i.e. for the background variability which interferes with the variability of the treatment represented by the standard deviation (SD). In all circumstances, the reader can easily estimate the SD from the values of SE by multiplying them by the square root of the “n”, which are shown for all the groups in the Figure 1. The mean ± SE presentation of the data is common for such studies and should not be considered as incorrect.
Reviewer 2 Report
Reviewer’s comments:
Overall, this is a well-written paper that presents interesting results on very relevant topic.
This paper appears to extend the authors’ previous work (accepted, but still unpublished) which showed H.alveilvei reduced food intake and body weight gain in WT mice fed a HFD and ob/ob mice. The current work combines these models, HFD ob/ob mice, and again the HA4597 strain of H.alveilvei suppressed food intake and body fat gain.
General comments:
There are some concerns regarding methods and statistics. It is somewhat unclear what the primary aim of the study is, and how the study design will address the aim. Authors use a standard mouse diet, rather than a control diet matched for micronutrients. This is a relatively short study (~5 weeks), in which the HFD did not exacerbate insulin resistance. Data is presented from all groups, ANOVAs are performed, and yet t-tests are performed to analyze each outcome. This should not be done. Use of a 2-way ANOVA is also questioned, based on the design. No main effects of interactions are mentioned anywhere in the text. Sample sizes are unclear, and should be listed for outcomes. In particular, body weight gain appears to be similar between HFD and HFD + H.alvei (although body composition does appear to be different).
Together, these issues could alter the interpretation of data, and should be addressed.
Specific comments:
Abstract – Please state the age and sex of mice. Line 33 – consider removal of ‘was associated with’
Intro –
Line 74 – The excessive hyperphagia and macronutrient characteristics of this diet do not closely reflect the human condition, and leptin deficiency is not common. Please consider carefully rewording.
Methods –
Line 100 – the sample sizes listed are inconsistent (54 vs 87?). Please clarify.
Line 123 – It is unclear how gavage volumes were individualized to body mass, yet the same H.alvei dose was given to all.
Food intake is a major outcome of the study, yet mice were group housed. Was average cage intake treated as an n=1 or on individual mouse basis? This is an important distinction which affects the statistics, but not specified in results.
Did authors compute feeding efficiency or kcal intake? These may be of interest.
Statistics –
More detail is needed, and possibly changes in analyses.
Results –
Line 171 – In Figure 2A from D-7 to D-1, does this not represent the SD group with sham treatment, and demonstrate stable body weight?
Figure 3A – It appears there are no differences until D15, is this true?
A repeated measures or cumulative weight gain are standard presentation of data. The AUC is considered less appropriate and may inflate results.
Line 244 – reference to t-test.
Figure 6 – please specify when the ad libitum samples were measured in text. It is somewhat surprising the values in (A) are equal to or higher the peak response to the OGTT. Is n=7/8 or 12 for each graph?
Line 274 – The increase in TG is not ‘suggesting insulin resistance’.
Overall, the presentation of data is confusing. It is difficult to determine what differences exist and when in most instances (especially Figures 5 & 6). In Figure 6 letters appear and are combined with *
Discussion –
The data demonstrating similar food intake between HFD and HFD + H.alvei after ~30 days warrants mentioning.
Glycemia – The lower glucose concentration observed in HFD + H.alvei could be impacted by reduced food (and CHO vs SD) intake. Some mention of this should be included.
Line 284 – In Figure 1, as denoted, the induction in obesity is apparent after 5 days (D-12 to D-7).
Line 286 – The HFD only results in increased total cholesterol and ALAT, please be specific, as there was no indication of altered glycemic response in the HFD group here.
Author Response
Response: We thank the Reviewer for the appreciation of our paper and for very useful suggestions. Below we have address all the general and specific comments.
Reviewer: It is somewhat unclear what the primary aim of the study is, and how the study design will address the aim.
Response: The primary aim of the study was stated in l. 72-73. “In the present study we aimed to test the anti-obesity efficacy of the H.alvei HA4597™, in a mouse model of obesity characterized by both hyperphagia and HFD-induced adiposity”. Further in l. 88-89, we explain how we will address this aim: “Thus, in the present study we used HFD to exacerbate obesity in hyperphagic ob/ob mice and then treated them chronically with H.alvei HA4597™ to compare its effects with untreated mice.” Moreover, the figure 1 illustrates in details the experimental design. In our view, this information is clear and sufficient to explain the aim of the study and its design.
Reviewer: Authors use a standard mouse diet, rather than a control diet matched for micronutrients.
Response: We are not sure if the Reviewer meant micro- or macro-nutrients in this comment. The standard diet was used to demonstrate the differences induced by HFD in adiposity and food intake. The data clearly show such differences.
Reviewer: This is a relatively short study (~5 weeks), in which the HFD did not exacerbate insulin resistance.
Response: We did not analyze response to insulin and cannot be sure about the level of insulin resistance, but an exacerbation of insulin resistance was not the aim of the present HFD model. As stated above, the aim of the model was a combination of hyperphagia and HFD-induced adiposity.
Reviewer: Data is presented from all groups, ANOVAs are performed, and yet t-tests are performed to analyze each outcome. This should not be done.
Response: We agree that using both multiple and individual group tests may look inappropriate in some circumstances. The purpose of doing so in our study is due to the nature of this exploratory preclinical analysis of the effect of H.alvei treatment in which a simple comparison between individual groups is interesting from the mechanistic point of view.
Reviewer: Use of a 2-way ANOVA is also questioned, based on the design.
Response: We agree with this comment and are sorry for misleading, in fact we did not performed 2-way ANOVA but a 2-way repeated measurement (RM) ANOVA which represents the best choice for this study when the dynamics of food intake or body weight changes are compared between different treatment groups. This mistake has been corrected in both the methods and the legends.
Reviewer: No main effects of interactions are mentioned anywhere in the text.
Response: As in the majority of the scientific publications in this research field, the results of the statistical analysis of our study report the standard outcomes. There is no formal requirement of this journal to present the main effects of interactions. As we stated above, this is an exploratory study which does not require the detailed presentation of the ANOVA table.
Reviewer: Sample sizes are unclear, and should be listed for outcomes.
Response: The sample sizes for each experimental group are shown in the Figure 1. To further clarify this question, the “n” was added to the figure legends.
Reviewer: In particular, body weight gain appears to be similar between HFD and HFD + H.alvei (although body composition does appear to be different).
Response: Indeed, the 2-way RM ANOVA results did not show significant differences between HFD and HFD+H.alvei groups. We modified a sentence in the result section to clarify this point.
Specific comments:
Abstract – Please state the age and sex of mice. Line 33 – consider removal of ‘was associated with’
Response: Age and sex were stated in the abstract. “was associated with” was replaced by “was accompanied by”.
Intro –Line 74 – The excessive hyperphagia and macronutrient characteristics of this diet do not closely reflect the human condition, and leptin deficiency is not common. Please consider carefully rewording.
Response: This sentence was modified to clarify the choice of the animal model in the present study and to mention the functional leptin resistance (not leptin deficiency) in obese humans.
Methods –Line 100 – the sample sizes listed are inconsistent (54 vs 87?). Please clarify.
Response: We are sorry for this mistake, the total number of mice was verified and corrected.
Line 123 – It is unclear how gavage volumes were individualized to body mass, yet the same H.alvei dose was given to all.
Response: The total volume was adjusted to body mass by 0.9% NaCl solution.
Food intake is a major outcome of the study, yet mice were group housed. Was average cage intake treated as an n=1 or on individual mouse basis? This is an important distinction which affects the statistics, but not specified in results.
Response: Food intake is reported per individual mouse, this was specified in the figure legends.
Did authors compute feeding efficiency or kcal intake? These may be of interest.
Response: During development of the combined model of hyperphagia and HFD we confirmed that both increased caloric and total food intake characterized our model as compared to the standard diet-fed ob/ob mice (Figure 2B,C). For the effects of H.alvei we report the total food intake since both treated and untreated groups had the same HFD. For the readers who is interested to estimate the caloric intake, the energy density of the diets have been included in the methods.
Statistics – More detail is needed, and possibly changes in analyses.
Response: The 2-way RM ANOVA question was clarified, see above.
Results – Line 171 – In Figure 2A from D-7 to D-1, does this not represent the SD group with sham treatment, and demonstrate stable body weight?
Response: Yes, it does.
Figure 3A – It appears there are no differences until D15, is this true?
Response: Yes, according to the results of 2-way RM ANOVA
A repeated measures or cumulative weight gain are standard presentation of data. The AUC is considered less appropriate and may inflate results.
Response: We agree that the AUC is a sensitive analysis that is why we present it side-by-side with the actual dynamics of body weight and body weight gain changes. Therefore, the reader can see if AUC may or may not “inflate” the results.
Line 244 – reference to t-test.
Response: The Figure 6B shows only the 2-way RM ANOVA analysis. The reference to the t-test, in the text is used to complete the statistical analysis for this exploratory study as discussed above.
Figure 6 – please specify when the ad libitum samples were measured in text. It is somewhat surprising the values in (A) are equal to or higher the peak response to the OGTT. Is n=7/8 or 12 for each graph?
Response: This question was clarified in the legend. We agree that in the ad libitum conditions, plasma glucose levels are somewhat elevated vs. OGTT, this may result from blood sampling during natural glucose peaks.
Line 274 – The increase in TG is not ‘suggesting insulin resistance’.
Response: We agree, although such association is common in type 2 diabetes, it is not causal. This suggestion was deleted.
Overall, the presentation of data is confusing. It is difficult to determine what differences exist and when in most instances (especially Figures 5 & 6). In Figure 6 letters appear and are combined with *
Response: We are sorry if the presentation of data may appear confusing, the way how it is presented was selected to avoid overloading of graphs. Thus, the visual presentation of data in graphs was combined with the description of statistical tests in the figure legends. Both should be consulted to avoid confusion.
Discussion – The data demonstrating similar food intake between HFD and HFD + H.alvei after ~30 days warrants mentioning.
Response: Yes, the difference between these 2 groups disappeared after 30 days, due to a progressive decrease of total food intake in HFD-group. We added a sentence to discuss this phenomenon.
Glycemia – The lower glucose concentration observed in HFD + H.alvei could be impacted by reduced food (and CHO vs SD) intake. Some mention of this should be included.
Response: We thank the reviewer for this suggestion, the corresponding mentions were added.
Line 284 – In Figure 1, as denoted, the induction in obesity is apparent after 5 days (D-12 to D-7).
Response: Yes, thank you for this comment, it was indeed 5 days, now corrected.
Line 286 – The HFD only results in increased total cholesterol and ALAT, please be specific, as there was no indication of altered glycemic response in the HFD group here.
Response: To avoid the non-specific allegations in this summary sentence, the reference for metabolic alteration was deleted.